# Integrative Metabolomics and Proteomics Detected Hepatotoxicity in Mice Associated with Alkaloids from *Eupatorium fortunei* Turcz.

**DOI:** 10.3390/toxins14110765

**Published:** 2022-11-05

**Authors:** Ke Zan, Wei Lei, Yaolei Li, Ying Wang, Lina Liu, Tiantian Zuo, Hongyu Jin, Shuangcheng Ma

**Affiliations:** 1National Institutes for Food and Drug Control, Beijing 102629, China; 2State Key Laboratory of Component-Based Chinese Medicine, Tianjin University of Traditional Chinese Medicine, Tianjin 301617, China

**Keywords:** *Eupatorium fortunei* Turcz., glycerophospholipid metabolism, hepatotoxic mechanisms, pyrrolizidine alkaloids

## Abstract

The traditional Chinese herbal medicine *Eupatorium fortunei* Turcz. (*E. fortunei*) has been widely adopted to treat nausea, diabetes, siriasis, and poor appetite. However, *E. fortunei* contains multiple pyrrolizidine alkaloids (PAs). This study aimed to investigate the hepatotoxicity of total alkaloids in *E. fortunei* (EFTAs) and identify the toxic mechanisms of EFTAs on hepatocytes. Liquid chromatography with a tandem mass spectrometry assay with reference standards indicated that EFTAs mainly consisted of eight PAs whose content accounted for 92.38% of EFTAs. EFTAs markedly decreased mouse body and liver weights and increased the contents of AST and ALT. The histopathological assays demonstrated that, after exposition to EFTAs, the structures of hepatocytes were damaged and the fibrosis and apoptosis in hepatocytes were accelerated. Moreover, EFTAs increased the serum level of inflammatory cytokines and aggravated circulating oxidative stress. A combination of hepatic proteomics and metabolomics was used to investigate the toxic mechanisms of EFTAs. The study revealed that EFTAs seriously disrupted glycerophospholipid metabolism by upregulating the contents of lysophosphatidylglycerol acyltransferase 1 and phosphatidylinositol and downregulating the contents of choline/ethanolamine kinase beta, choline-ethanolamine phosphotransferase 1, phospholipase D4, 1-acylglycerophosphocholine, phosphatidylcholine, and dihydroxyacetone phosphate in the liver, resulting in detrimental inflammation, fibrosis, and apoptosis. This study revealed that EFTAs induced severe hepatotoxicity by disrupting glycerophospholipid metabolism.

## 1. Introduction

*Eupatorium fortunei* Turcz. (in Chinese, Pei-Lan; *E. fortunei*) belongs to the Asteraceae family and is widely distributed in China, Korea, and Japan. *E. fortunei* expresses the traditional activities of resolving dampness, enlivening the spleen, promoting appetite, and releasing heat from the body in summer. It has been used as a traditional Chinese medicine and is documented in Chinese pharmacopeia [1]. Modern pharmacological studies have revealed that *E. fortunei* extracts induce various biological actions such as anti-inflammation, antibiosis, antiviral, antioxidation, and others [2,3,4]. Therefore, *E. fortunei* extracts have been widely used in Chinese herb compounds to treat nausea, diabetes, siriasis, poor appetite, and so on [5]. Furthermore, *E. fortunei* has also been adopted for treating severe acute respiratory syndrome coronavirus 2 infections; it also successfully ameliorates pulmonary symptoms in patients with COVID-19 infections [6].

Many phytochemical studies reported that *E. fortunei* contained multiple pyrrolizidine alkaloids (PAs) that were naturally occurring toxic compounds with genotoxic, neurotoxic, and hepatotoxic properties [4,7]. The Committee on Herbal Medicinal Products of the European Medicines Agency (EMA/HMPC) published the EMA/HMPC/893108/2011 Rev. 1 guidelines recommending that an adult should take less than 1.0 μg PAs through herbal medicinal intake every day. Hence, despite *E. fortunei* being beneficial to human health, we should be cautious in its use considering the toxic properties of total alkaloids present in it.

The main objective of the present study was to identify the potential toxic mechanisms of hepatotoxicity associated with EFTAs in vivo. First, the compositions of EFTAs were identified, and their contents were quantified using a Waters TQS mass spectrometer coupled with a Waters H-class UHPLC system. After C3H mice were exposed to EFTAs for 4 weeks, the body weight, liver weight, pathological histology, and biochemical factor assays were performed to assess the hepatotoxicity of EFTAs. A combination of hepatic proteomics and hepatic metabolomics techniques was used to identify the toxic mechanisms of EFTAs on hepatocytes. The results of this study revealed that EFTAs generated serious hepatotoxicity mainly by interfering with glycerophospholipid metabolism, oxidative phosphorylation, glutathione metabolism, and primary bile acid biosynthesis.

## 2. Results

### 2.1. PAs Dominated EFTAs

EFTAs were identified using liquid chromatography–mass spectrometry (LC–MS) with standards such as the retention time and characteristic ions (Appendix A), and then extracted and purified. Eight main PAs could be detected, including intermedine (Im), lycopsamine (Ly), rinderine (Rd), echinatine (En), rinderine N-oxide (RdNO), echinatine N-oxide (EnNO), intermedine N-oxide (ImNO), and lycopsamine N-oxide (LyNO) (Figure 1A,B). The contents of these eight PAs in the EFTA extract were determined. In the EFTAs, the contents of Im, Ly, Rd, RdNO, and ImNO were between 4% and 9%, while the contents of En and LyNO were between 14% and 18% (Table 1). The most abundant ingredient was EnNO, and its content was approximately 29.55% (Table 1). The content of eight PAs by weight accounted for 92.38% EFTAs. Among these eight PAs, En, RdNO, and EnNO could be detected in this herb for the first time.

### 2.2. EFTAs Induced Severe Hepatotoxicity

The heart, liver, spleen, lungs, and kidneys were separated from mice. EFTA exposition of 25 mg/(kg·d) for 4 weeks produced little change in the appearance of various organs (Figure 2A). EFTA exposition significantly decreased the mouse body and liver weight (Figure 2B,C). The levels of aspartate transaminase (AST) and alanine transaminase (ALT), the pivotal biomarkers of hepatic function, were markedly elevated compared with control mice, which indicated that EFTA exposition adversely affected hepatic functions (Figure 2D,E).

Hematoxylin and eosin (H&E) staining was performed on liver sections to assess the structural injuries of hepatocytes stimulated by EFTAs. Obvious liver lesions that increased cytoplasmic volume, enlarged nuclei, and distended cells with vacuolated cytoplasm were observed in mice exposed to EFTAs for 4 weeks (Figure 2F). The content of collagens in the liver was augmented in the EFTA group compared with the control group, which suggested that EFTA exposition induced extensive hepatic fibrosis (Figure 2G,H). Furthermore, EFTA exposition for 4 weeks significantly increased apoptotic hepatocytes, suggesting that EFTAs damaged the normal structures of hepatocytes and accelerated their apoptosis (Figure 2I,J).

### 2.3. EFTAs Exacerbated Circulating Inflammatory and Oxidative Stress

The serum levels of inflammatory cytokines, interleukin (IL)-6 (107.97 ± 5.58 vs. 126.54 ± 5.32), tumor necrosis factor (TNF)-α (742.08 ± 47.56 vs. 870.12 ± 41.83), nuclear factor kappa B (NF-κB) (1173.95 ± 56.36 vs. 1305.15 ± 69.60), and inducible nitric oxide synthase (iNOS) (19.21 ± 0.92 vs. 24.77 ± 0.84) were elevated in mice exposed to EFTAs compared to control mice (Figure 3A–D). The activities of glutathione S-transferase (GST) (28.71 ± 3.93 vs. 40.67 ± 6.11, control vs. EFTA) and lactate dehydrogenase (LDH) (1457.65 ± 141.97 vs. 1806.47 ± 178.29, control vs. EFTA) increased in the mouse serum exposed to EFTAs compared to control mice, indicating that the hepatocyte was ruptured and the two aforementioned proteins leaked out (Figure 3E,H). The activity of superoxide dismutase (SOD) (152.70 ± 9.04 vs. 138.95 ± 12.55) decreased in the mouse serum exposed to EFTAs compared to control mice (Figure 3F). The ratio of glutathione disulfide (GSSG) to reduced glutathione (GSH) (0.26 ± 0.052 vs. 0.94 ± 0.16) increased enormously in mice exposed to EFTAs compared to control mice, indicating the stimulation of oxidative stress by EFTAs (Figure 3G).

### 2.4. Hepatic Proteomics Revealed That EFTAs Disturbed Oxidative Phosphorylation and Glycerophospholipid Metabolism

Hepatic proteomics was carried out to uncover the potential toxic mechanisms of EFPAs in liver cells. The hepatic proteomics identified 92 differentially expressed proteins (DEPs), including 41 upregulated DEPs and 51 downregulated DEPs (Figure 4A and Appendix A). When the upregulated DEPs were submitted to the Gene Ontology (GO) and Kyoto Encyclopedia of Genes and Genomes (KEGG) databases, the primary bile acid biosynthesis, cholesterol metabolism, and oxidative phosphorylation were identified as important pathways disturbed by EFTAs with the most enrichment factors, while oxidative phosphorylation signaling acquired the lowest *p* values (Figure 4B). After analysis, the upregulated MT-ATP8, NDUFC2, LHPP, and NDUFA7 and the downregulated ATP6V1E1 were enriched in the oxidative phosphorylation signaling pathway, indicating that oxidative phosphorylation was perturbed by EFTAs (Figure 4C).

When the downregulated DEPs were subjected to GO and KEGG analysis, the ether lipid, glycerophospholipid, and arachidonic acid metabolisms were identified as significant pathways disturbed by EFPAs with the most enrichment factors (Figure 4D). Among these three pathways, glycerophospholipid metabolism played a vital role in protecting hepatic functions, and the upregulated lysophosphatidylglycerol acyltransferase 1 (LPGAT1) and downregulated choline/ethanolamine kinase beta (CHKB), EPT, and phospholipase D4 (PLD4) were enriched in the glycerophospholipid metabolism signaling pathway (Figure 4E). When the upregulated and downregulated DEPs were all put into the STRING database for protein–protein interaction analysis, oxidative phosphorylation and glycerophospholipid metabolisms were also found to be outstanding pathways disturbed by EFTAs with false discovery rates of 0.017 and 0.036, respectively (Figure 4F).

### 2.5. Hepatic Metabolomics Demonstrated That EFTAs Disturbed Glutathione Metabolism, Primary Bile Acid Biosynthesis, and Glycerophospholipid Metabolism

Hepatic metabolic profiles were detected and analyzed in ESI positive and negative modes to discover the metabolic interference of EFTAs. The OPLS-DA score plots demonstrated that mouse hepatic metabolic profiles exposed to EFTAs deviated from the control mice in both positive [intercepts: R2 = (0.0, 0.993), Q2 = (0.0, 0.267)] and negative [intercepts: R2 = (0.0, 0.96), Q2 = (0.0, –0.048)] modes (Figure 5A,C and Appendix A). Furthermore, the S plots and the variable influence in the projection (VIP) values were applied to identify differential metabolites that contributed to the separation of hepatic metabolic profiles in control and EFTA mice, and 43 differential metabolites were identified (Figure 5B,D and Appendix A). When the differential metabolites were submitted to the MetaboAnalyst platform, the primary metabolic pathways were identified as glutathione metabolism, primary bile acid biosynthesis, and glycerophospholipid metabolism, while glutathione metabolism signaling included D-glutamine and D-glutamate metabolism pathways (Figure 5E,F).

### 2.6. EFTAs Disturbed Glycerophospholipid Metabolism

A comprehensive analysis combining DEPs with differential metabolites was also carried out using MetaboAnalyst 5.0. The results demonstrated that the glycerophospholipid metabolism pathway was the key biosignaling pathway disturbed by EFTAs, with the highest pathway impact and lowest *p* values (Figure 6A). Western blots were used to assess the expressions of CHKB, PLD4, choline-ethanolamine phosphotransferase 1 (CEPT1), and LPGAT1 to validate this hypothesis, which were enriched in the glycerophospholipid metabolism pathway using hepatic proteomics. The results of Western blots revealed that the expression of LPGAT1 was upregulated in the three independent samples from the EFTA group compared with that from the control group, while the expression of CHKB, CEPT1, and PLD4 was slightly reduced in mouse livers exposed to EFTAs, reflecting similar trends in the results obtained via hepatic proteomics (Figure 6B).

## 3. Discussion

PAs and their N-oxides are naturally occurring toxic secondary metabolites in plants, and the intake of a large amount of PAs and their N-oxides causes hepatic sinusoidal obstruction syndrome. Long-term intake of a small amount of PAs can also lead to liver fibrosis, cancer, and other liver complications [8,9]. *E. fortunei* is a traditional Chinese medicine with many beneficial active compounds such as terpenoids, flavonoids, phenylpropanoids, and essential oils [10]. However, this study revealed that *E. fortunei* also contained PAs and their N-oxides, accounting for 92.38% of EFTAs. PAs are metabolized by the CYP3A4 isozyme to produce reactive dehydropyrrolizidine alkaloids (DHPAs), which can be further hydrolyzed to (±)-6,7-dihydro-7-hydroxy-1-hydroxymethyl-5*H*-pyrrolizine (DHP) [11]. Both DHPAs and DHP can bind to biologically functional molecules generating hepatotoxicity. As PAs comprise a major portion of EFTAs, the hepatotoxicity due to EFTAs is not insignificant. Considering the wider applications of *E. fortunei* in herbal treatments, further studies are needed to investigate the toxic mechanisms and hepatotoxicity associated with EFTAs to ensure the rational use of *E. fortunei*.

This study revealed that the body and liver weights markedly decreased in mice exposed to EFTAs for 4 weeks, while the levels of biomarkers of liver damage, AST and ALT, significantly increased. Furthermore, the histopathological assays demonstrated that after EFTA exposition, the structure of hepatocytes was damaged, and the fibrosis and apoptosis of hepatocytes were accelerated. The results of this study confirmed that EFTAs did cause obvious hepatotoxicity.

Hepatic proteomics combined with hepatic metabolomics was conducted to reveal the toxic mechanisms of EFTAs. A previous study reported that PAs induced uncoupling of oxidative phosphorylation, disturbed energy metabolism, and resulted in the accumulation of oxygen-free radicals [12]. Consistent with the results of this study, EFTA exposition increased the circulating oxidative stress, and hepatic proteomics helped indicate that EFTAs disturbed oxidative phosphorylation by upregulating the expression of MT-ATP8, NDUFC2, LHPP, and NDUFA7 and downregulating the expression of ATP6V1E1. Moreover, glutathione metabolism modulated the detoxification of free radicals and toxic oxygen radicals, protecting hepatocytes from toxic oxygen and electrophilic metabolites [13,14]. The results of hepatic metabolomics revealed that EFTAs disrupted glutathione metabolism, which also boosted oxidative stress. Bile acid biosynthesis signaling regulated the absorption, transformation, and secretion of lipids, which closely modulated the early outcomes of autoimmune hepatitis [15,16]. Glycerophospholipids are a type of structural building block for cytomembrane integrity and energy stores, and they participate in many cell signaling processes including inflammatory progress, immune response, and liver fibrosis [17,18]. In this study, the results of hepatic proteomics and metabolomics both indicated that EFTAs seriously disrupted glycerophospholipid metabolism by upregulating the contents of LPGAT1 and phosphatidylinositol and downregulating CHKB, CEPT1, PLD4, 1-acylglycerophosphocholine, phosphatidylcholine, and dihydroxyacetone phosphate (Figure 6C). The disturbed glycerophospholipid metabolism accompanied by hepatocyte fibrosis and apoptosis was observed in mice exposed to EFTAs. CHKB catalyzed the first step during the synthesis of the membrane phospholipid phosphatidylcholine, and CHKB defects led to an obvious inability to use fatty acids for β-oxidation, resulting in the accumulation of extensive fatty acids [19]. Therefore, EFTAs reduced the expression of CHKB, which destroyed the integrity of the hepatocyte membrane and accumulated extensive fatty acids in the hepatocyte. CEPT1 regulated the de novo phospholipogenesis, which was indispensable for phospholipid activation of transcription factors that promoted tissue recovery and endothelial protection in the liver. The decrease in CEPT1 induced by EFTAs undoubtedly disrupted the de novo phospholipogenesis and generated tissue and endothelial defects in the liver [20]. PLD4 not only participated in the glycerophospholipid metabolism, regulating mitochondrial function but also modulated inflammatory signaling induced by endosomal TLR and cytoplasmic/STING nucleic acid sensing pathways [21]. Furthermore, LPGAT1 also participated in mitochondrial DNA depletion, mitochondrial dysfunction, and oxidative stress [22]. The results of this study revealed that EFTAs suppressed the PLD4 expression and elevated the level of LPGAT1 in the mouse liver, suggesting that mitochondrial dysfunction and inflammation occurred in the mouse liver exposed to EFTAs. This study also revealed that EFTAs induced severe hepatotoxicity by disrupting glycerophospholipid metabolism, reminding us that patients using *E. fortunei*-containing prescriptions should be paid serious attention.

## 4. Conclusions

In this study, we investigated the hepatotoxicity associated with EFTAs and identified toxic mechanisms of EFTAs in hepatocytes. *E. fortunei* mainly consisted of eight PAs whose contents accounted for 92.38% of EFTAs. EFTAs induced severe hepatotoxicity by aggravating inflammatory and oxidative stress and exacerbating the fibrosis and apoptosis of the hepatocyte. Hepatic proteomics combined with metabolomics indicated that EFTAs disturbed glycerophospholipid metabolism, oxidative phosphorylation, glutathione metabolism, and primary bile acid biosynthesis. The daily intake of *E. fortunei* recommended by doctors is between 5 and 10 g for an adult, and the daily recommended intake of PAs by the EMA/HMPC is less than 1.0 μg. However, the doctor-recommended amount of herbs contains PAs from 4.92 to 9.85 μg, which far exceeds the PA intake limit recommended by EMA/HMPC. Therefore, long-term use of *E. fortunei* will elicit liver damage, but short-term consumption of *E. fortunei* should also be paid serious attention by practitioners. Moreover, anti-inflammatory and antioxidative agents can be used to cure glycerophospholipid metabolic disorders, which would help heal the liver damage induced by *E. fortunei*.

## 5. Materials and Methods

### 5.1. Chemicals and Reagents

The ALT and AST kits were bought from Sigma–Aldrich (St. Louis, MO, USA). The LDH and enzyme-linked immunosorbent assay (ELISA) kits for mouse IL-6, TNF-α, NF-κB, and iNOS were purchased from Jiancheng Bioengineering Institute (Nanjing, China). The GST, SOD, and GSSG/GSH kits were purchased from Abcam (BCG, Boston, MA, USA). A bicinchoninic acid (BCA) kit was obtained from Beyotime Biotechnology (Shanghai, China). The primary antibodies against LPGAT1 (Cat# orb184994), PLD4 (Cat# bs-12718R), and CEPT1 (Cat# orb156344) were bought from Biorbyt Ltd. (Cambridge, UK). The primary antibody against CHKB (Cat# PA5-112385) was obtained from Invitrogen (Thermo Fisher Scientific, Carlsbad, MA, USA). The primary antibody against glyceraldehyde-3-phosphate dehydrogenase (Cat# BA2913) was acquired from Boster Biological Technology Co. Ltd. (Wuhan, China). The Horseradish peroxidase (HRP)-conjugated secondary antibodies (Cat# bs-40295G-HRP) were purchased from Bioss, Inc. (Beijing, China). The standards Im, Ly, En, EnNO, and RdNO were acquired from PhytoLab (Vestenbergsgreuth, Germany). The standards Rd, ImNO, and LyNO were bought from Shanghai Zzbio Co., Ltd. (Shanghai, China). All other reagents and chemicals used in this study were of analytical grade.

### 5.2. Plant Samples

*E. fortunei* herbal samples were gathered from Changde (Changde, China) in August 2021. The herbal samples were authenticated by Dr Ke Zan (National Institutes for Food and Drug Control, Beijing, China), and the voucher specimens (No. PL-2021-01) were stored at the National Institutes for Food and Drug Control.

### 5.3. Preparation of Total Alkaloids from E. fortunei

Dried herbs (30 kg) of the aboveground part of *E. fortunei* were cut and extracted via refluxing 10 times with a volume of 50% (*v*/*v*) ethanol for 2 h. After the ethanol was removed under reduced pressure, the pH of the extracted solution was adjusted to approximately 2.5 by adding 2 mol/L of sulfuric acid.

The aforementioned acidified extract was subjected to cation exchange resin open-column chromatography (20 × 100 cm^2^ and eluted using 50 L of CH_3_OH and 50 L of NH_4_OH-CH_3_OH (1:3, *v*/*v*) to yield two fractions (1 and 2). Fraction 2 was further refined using octadecylsilyl open-column chromatography (6 × 70 cm^2^) (MeOH/H_2_O, 10:90 to 50:50, *v*/*v*) to afford subfractions 1 and 2. Total alkaloid extracts (32 g) were obtained by removing the solvent in subfraction 2. The contents of eight PAs in the extract were identified and determined using a Waters TQS Triple Quad mass spectrometer coupled with a Waters H-class UHPLC system (Waters Corporation, Milford, MA, USA) via the method established in our previous study [23].

### 5.4. Animals

Twenty male C3H mice were bought from Beijing Vital River Laboratory Animal Technology Co., Ltd. (Lot No. 110011220103268358, Beijing China), aged 42–55 days. All mice were fed water and food ad libitum and accommodated to 24 °C ± 1 °C, 40–60% humidity, and 12 h light/dark cycles. Before the experiment, all mice were adapted to the environment for 7 days.

Twenty male C3H mice were randomly divided into two groups, either the control group (*n* = 10) or the EFTA exposure group (*n* = 10). The EFTA exposure model was established by intragastrically treating mice with 25 mg/(kg·d) EFTAs, while control mice were administrated equivalent physiological saline orally. The animal experiments lasted 4 weeks and body weights were recorded. After treatment for 28 days, the whole-blood samples were acquired from mice’s eyes and the serum samples were separated from whole blood following the settlement for 30 min at room temperature. The heart, liver, spleen, lungs, and kidneys were separated from the mice and weighed. The serum samples and liver tissues were stored at −80 °C for the following tests. Each liver tissue was divided into three parts, and the same parts of 8 liver tissues from every group were used for pathological histology assays. Another part of the 8 liver tissues in each group was applied for hepatic metabolomics and the final part of the liver tissues in every group were adopted for hepatic proteomics and Western blots assay.

### 5.5. Pathological Histology and TUNEL Staining

The liver tissues were washed using phosphate buffer saline (PBS) and slightly drained with a paper towel. Subsequently, the liver tissues were fixed in ice-cold formalin fixative (4% formaldehyde in PBS) for 12 h. The prefixed liver tissues were embedded in paraffin and sectioned into 5 μm thick specimens. The liver sections were stained using H&E and Masson stains. TUNEL staining was conducted following the manufacturer’s protocols. The images of stained sections were captured using a light microscope CKX41 (Olympus, Tokyo, Japan), and the data were quantified using Image J software (NIH, Bethesda, MD, USA).

### 5.6. Biochemical Assay

The contents of inflammatory cytokines IL-6, TNF-α, NF-κB, and iNOS and the liver functional biomarkers ALT and AST in the serum samples were measured using commercial kits following the manufacturer’s protocols. The GSSG/GSH ratios and the activities of GST, SOD, and LDH were detected using commercial kits following the manufacturer’s protocols. The absorbance values were acquired via a SPARK microplate reader (TECAN, Männedorf, Switzerland).

### 5.7. Hepatic Proteomics

#### 5.7.1. Protein Sample Extraction

The liver tissues were pulverized in liquid nitrogen and subsequently lysed in ice-cold lysis buffer (8 mol/L urea with 1% protease inhibitor cocktail) for 30 min. The liver homogenates were centrifuged at 12,000 rpm and 4 °C for 15 min, and the protein concentration of supernatants was detected using a BCA kit following the manufacturer’s protocols. Proteins (200 µg) were first digested by trypsin at 37 °C for 4 h and then by pancreatin at 37 °C for 16 h. The tryptic peptides were collected via centrifugation at 12,000 rpm and 4 °C for 10 min.

#### 5.7.2. LC–MS Assay for Proteomics

The tryptic peptides were fractionated into 60 parts using a Thermo Acclaim PepMap reversed-phase liquid chromatography (RPLC) C18 column (300 µm × 5 mm, 5 µm) with a gradient elution of 6–35% acetonitrile (pH 9.0) for more than 100 min. Then, these fractions were dried and redissolved by an eluent using a Thermo Acclaim PepMap RPLC C18 column (75 µm × 150 mm, 3 µm) in a Thermo ultra-performance liquid chromatography (UPLC) system (Thermo Fisher Scientific, Carlsbad, MA, USA) with a flow rate of 0.3 µL/min. The optimal mobile phase was constituted by a linear gradient system of (A) 0.1% formic acid in 2% acetonitrile and (B) 0.1% formic acid in 80% acetonitrile: 0–8 min, B 6–9%; 8–35 min, B 9–14%; 35–75 min, B 14–30%; 75–96 min, B 30–40%; 96–100 min, B 40–90%.

MS spectra were obtained via Q-Exactive HF mass spectrometry (Thermo Fisher Scientific, Carlsbad, MA, USA). The optimal conditions were as follows: The mass range for the full scan was set at 300–1400 m/z; the resolution of peptides was 120,000 for the MS scan and 175,00 for the MS/MS scan; the AGC target for the MS scan was 3e6 and 1e5 for the MS/MS scan; the maximum IT was 40 ms for the MS scan and 60 ms for the MS/MS scan; TopN was 20 and NCE/stepped NCE was 27. The change in the DEP content was set to be more than 1.2-fold with *p* < 0.05 (Student’s *t* test, assuming unequal variance) and applied to compare proteins between the EFTA and control groups.

The GO (RRID: SCR_007,691), KEGG (http://www.kegg.jp/, accessed on 10 May 2022), and STRING 11.0 (https://string-db.org/, accessed on 12 May 2022) databases were adopted for protein classification and protein functional enrichment using a false discovery rate of ≤0.05.

### 5.8. Hepatic Metabolomics

#### 5.8.1. Hepatic Metabolite Extraction

The hepatic samples (*n* = 8) were thawed at 4 °C and 20 mg samples were ground to homogenates followed by centrifugation at 3000 rpm and 4 °C for 10 min. A 100-µL aliquot of the supernatant was mixed with 400 µL of 70% methanol for 15 min to precipitate the proteins. The mixture was centrifuged at 12,000 rpm and 4 °C for 10 min, and 200 µL of the supernatant was subjected to LC–MS analysis. The quality control sample was prepared by mixing an equal amount (20 µL) of the aliquot from each sample.

#### 5.8.2. LC–MS Assay for Metabolomics

The LC–MS analysis was conducted using an Agilent 1290 UPLC system coupled with a 6545 quadrupole TOF MS system equipped with an Agilent JetStream ESI interface operated via Masshunter Workstation software B.04.01 (Agilent Technologies, Santa Clara, CA, USA). Metabolite separation was performed using an ACQUITY UPLC HSS T3 column (2.1 mm × 100 mm × 1.8 µm; Waters Corporation, Milford, MA, USA). The column was maintained at 30 °C, and the flow rate was 0.4 mL/min. A 2 µL aliquot of each sample was injected. The optimal mobile phase consisted of a linear gradient system of (A) 0.1% formic acid in water and (B) 0.1% formic acid in acetonitrile: 0–11.0 min, 5–90% B; 11.0–12.0 min, 90% B; 12.0–12.1 min, 90–5% B, 12.1–14.0 min, 5% B. The parameters applied for the mass detection were as follows: Gas temperature of 325 °C; gas flow of 8 L/min; nebulizer of 40 psi; sheath gas temperature of 325 °C; sheath gas flow of 11 L/min; capillary voltage of 2500 (positive) and 1500 (negative); capillary outlet voltage of 135 V; and collision energy of 30 V.

The raw MS data were converted into mzML format using ProteoWizard (http://proteowizard.sourceforge.net/tools.shtml, accessed on 9 May 2022) and pretreated using XCMS (http://www.bioconductor.org/packages/release/bioc/html/xcms.html accessed on 9 May 2022). An unsupervised PCA model was applied to assess the quality, homogeneity, outlier identification, and dominant trends of the group separation. A supervised OPLS-DA was employed to discriminate the classes and identify differential variables. The differential variables with VIP values of >1.5 were identified as biomarkers for EFTA detection. The differential metabolites were derived by searching the exact molecular mass data from the redundant m/z peaks against the online HMDB (http://www.hmdb.ca/, accessed on 12 May 2022), METLIN (http://metlin.scripps.edu/, accessed on 12 May 2022), and KEGG databases. The metabolic pathway analysis and the unsupervised heatmap with a Euclidean distance measure were conducted using MetaboAnalyst 5.0 (http://www.metaboanalyst.ca/, accessed on 13 May 2022).

### 5.9. Western Blots

The protein extraction process was the same as the procedures used in Section 5.7.1. The aliquots of protein samples were subjected to SDS-PAGE (12%) for separation and then transferred onto PVDF membranes. The PVDF membranes loaded with proteins were first blocked using 5% skim milk for 1 h at 37 °C and then incubated with primary antibodies against LPGAT, PLD4, CEPT1, and CHKB overnight at 4 °C. The membranes were washed with PBS with Tween20 and then incubated with HRP-conjugated secondary antibodies at 37 °C for 1 h. The protein bands were developed using chemiluminescent HEP substrates and imaged using a Bio-Rad ChemiDocTM MP Imaging System (Bio-Rad, Hercules, CA, USA). The relative intensities of proteins were calculated using Image J software.

### 5.10. Statistical Analysis

The results of this study were expressed as the mean ± SD. Significant differences between the two groups were determined via the Mann–Whitney test in the pathological histology assay and biochemical assay, and *p* ≤ 0.05 indicated a statistically significant difference. Significant differences between the control and EFTAs groups in the Western blot assay were determined by a *t* test (and nonparametric tests), and *p* ≤ 0.05 indicated a statistically significant difference.

## Figures and Tables

**Figure 1 toxins-14-00765-f001:**
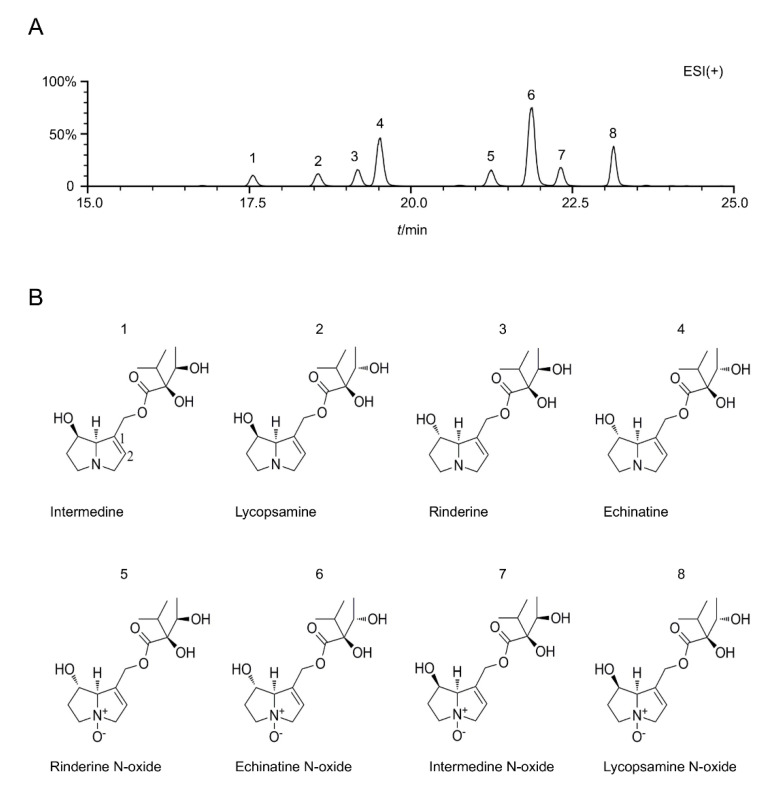
Chemical composition of alkaloid-enriched fraction of *E. fortunei* extract. (**A**) Total ion chromatogram of EFTAs in the positive ESI mode. (**B**) Chemical structures of PAs from *E. fortunei*.

**Figure 2 toxins-14-00765-f002:**
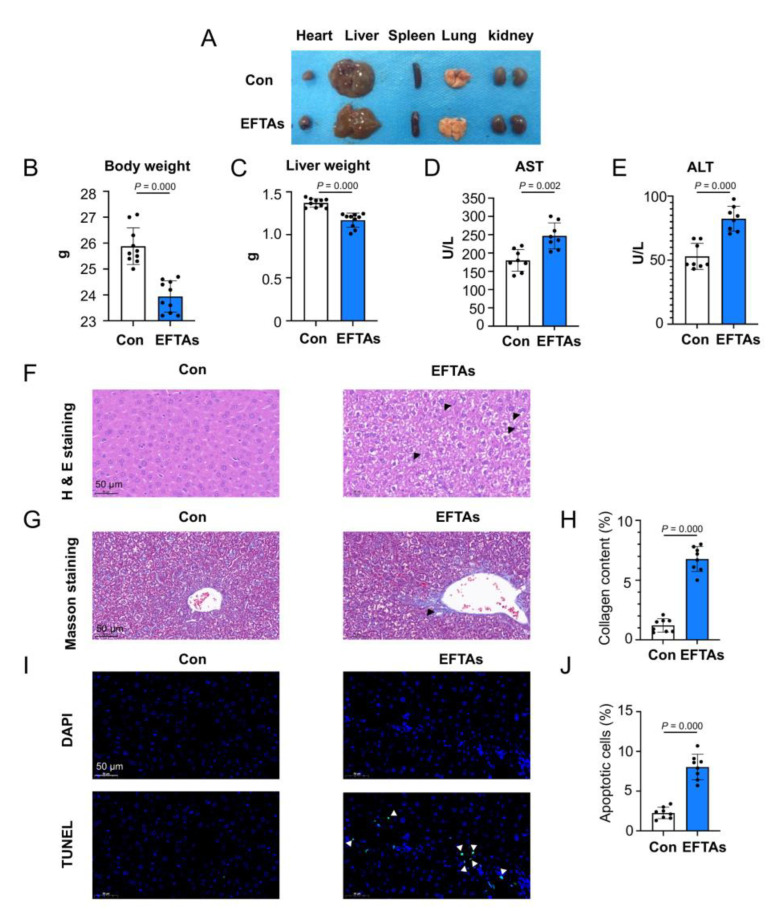
EFTAs induced severe hepatotoxicity. (**A**) Representative tissue states of the heart, liver, spleen, lungs, and kidneys from control and EFPA mice. Range of body weights (**B**) and liver weights (**C**) of control and EFPA mice (*n* = 10). AST activity (**D**) and ALT activity (**E**) in the serum samples of control and EFPA mice (*n* = 8). (**F**) Representative histological photomicrographs of mouse liver sections stained using H&E. The swollen cells were labeled by black triangle. (**G**,**H**) Representative photomicrographs of mouse slices stained using Masson’s trichrome, and the quantitative analysis of the collagen content (*n* = 8). The collagen fibers were stained as blue fibers and labeled by black triangle. (**I**,**J**) Representative photomicrographs of terminal deoxynucleotidyl transferase dUTP nick end labeling (TUNEL) assay and quantitative analysis of apoptotic liver cells (*n* = 8). The cell nucleus was stained by DAPI and presented as light blue dots. The apoptotic cells exhibited green light, being labeled by white triangle. Scale bar = 50 μm. Each bar represents the mean ± standard deviation (SD). *p* ≤ 0.05, indicating statistically significant differences.

**Figure 3 toxins-14-00765-f003:**
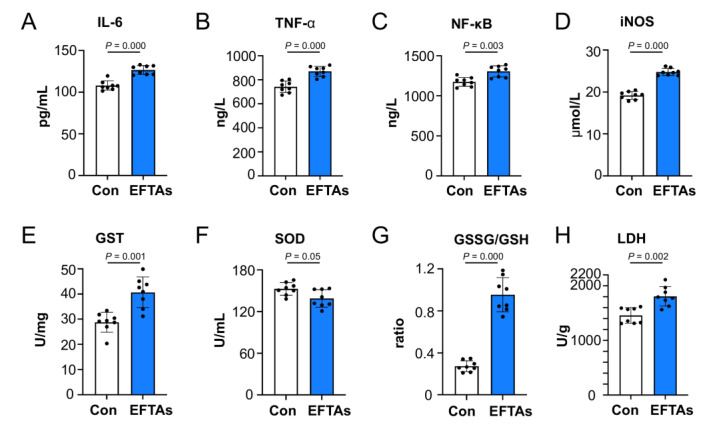
Influence of EFTAs on serum biochemical indicators in C3H mice. (**A**) IL-6; (**B**) TNF-α; (**C**) NF-κB; (**D**) iNOS; (**E**) GST; (**F**) SOD; (**G**) GSSG/GSH; and (**H**) LDH. Each bar represents the mean ± SD; *p* ≤ 0.05, indicating the statistically significant differences; *n* = 8.

**Figure 4 toxins-14-00765-f004:**
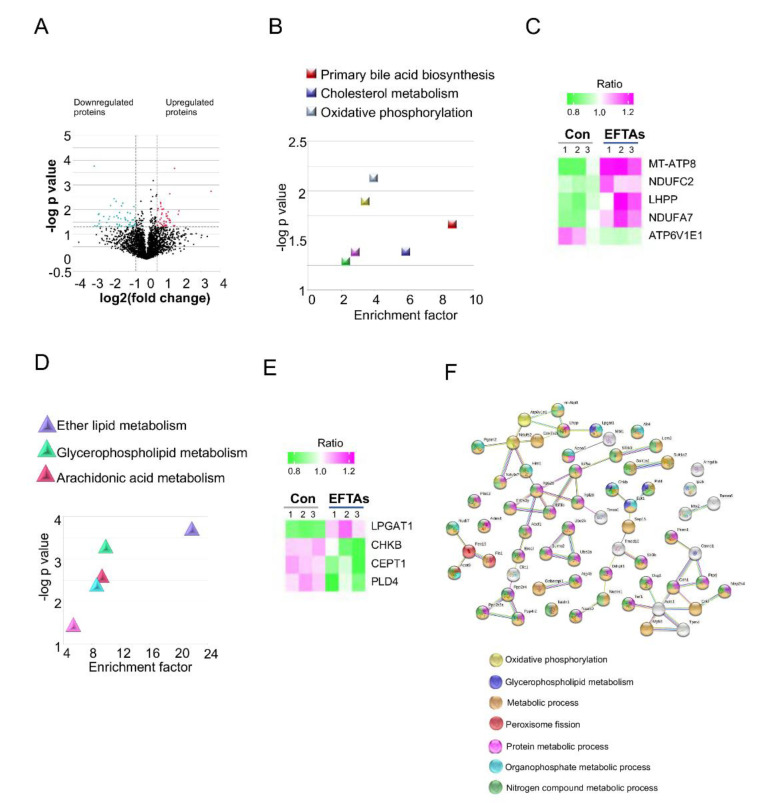
Hepatic proteomics revealed that EFTAs upregulated oxidative phosphorylation and downregulated glycerophospholipid metabolism. (**A**) Volcano plot of hepatic proteomics. Red scatters are upregulated proteins and green scatters are downregulated proteins. (**B**) Primary pathways that upregulated proteins were enriched in. The yellow, purple and green squares with low enrichment factors were not considered significantly. (**C**) Heatmap of upregulated proteins involved in oxidative phosphorylation signaling. (**D**) Primary pathways that downregulated proteins were enriched in. The light blue and pink triangles with low enrichment factors and p value were considered as insignificant pathways (**E**) Heatmap of downregulated proteins involved in glycerophospholipid metabolism signaling. (**F**) Protein–protein interaction of total DEPs in hepatic proteomics.

**Figure 5 toxins-14-00765-f005:**
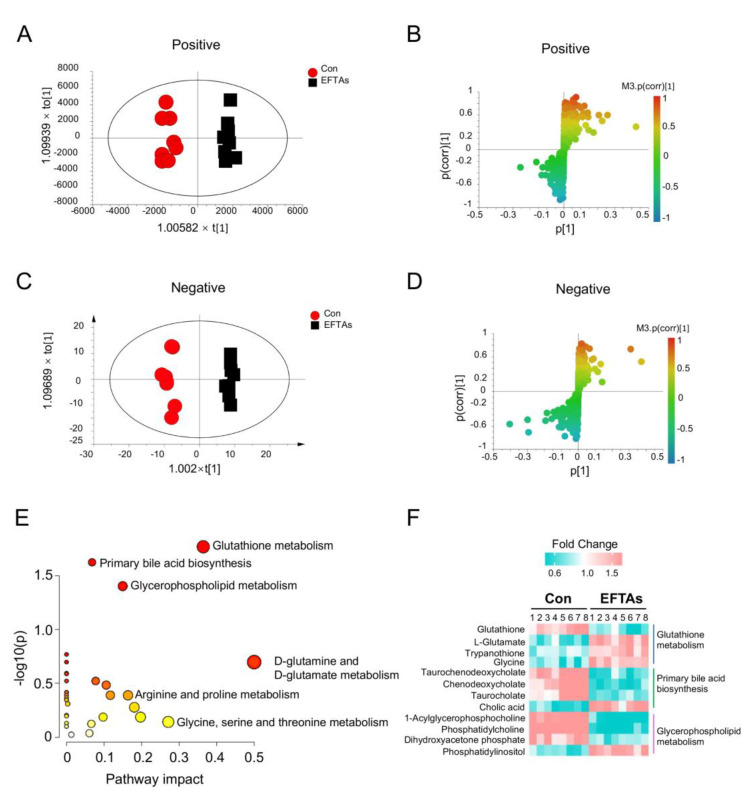
Hepatic metabolomics demonstrated that EFTAs modulated glutathione metabolism, primary bile acid biosynthesis, and glycerophospholipid metabolism. OPLS-DA plot (**A**) and S plot (**B**) of global hepatic metabolism in the positive ESI mode. OPLS-DA plot (**C**) and S plot (**D**) of global hepatic metabolism in the negative ESI mode. (**E**) Bubble plot of hepatic metabolomics. (**F**) Heatmap of differential metabolites enriched in glutathione metabolism, primary bile acid biosynthesis, and glycerophospholipid metabolism.

**Figure 6 toxins-14-00765-f006:**
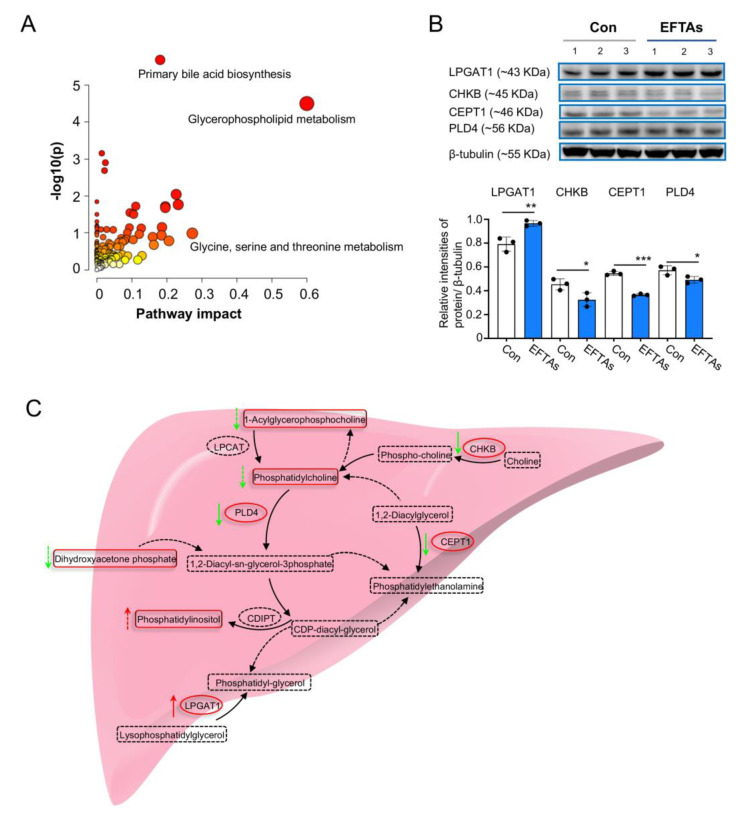
EFTAs primarily regulated glycerophospholipid metabolism in the mouse liver. (**A**) Bubble plot indicating primary signaling pathways involving both DEPs and differential metabolites. (**B**) Western blots of DEPs enriched in the glycerophospholipid metabolism pathway and their relative intensities in control and EFTAs groups. Each bar represents the mean ± SD; * *p* < 0.05, ** *p* < 0.01, *** *p* < 0.001 compared with the Con group, *n* = 3. (**C**) Dominant metabolic processes disturbed by EFTAs. The green arrow presents downregulation trend and the red arrow presents upregulation trend.

**Table 1 toxins-14-00765-t001:** Content of PAs in the EFTA extract.

Compound	Im	Ly	Rd	En	RdNO	EnNO	ImNO	LyNO	Total
Content (%)	4.15	4.77	6.39	17.28	7.14	29.55	8.63	14.47	92.38

## Data Availability

All data is contained within the article and Appendix A.

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
