# Peer review of "Integrative Metabolomics and Proteomics Detected Hepatotoxicity in Mice Associated with Alkaloids from Eupatorium fortunei Turcz."

_toxins, 2022, doi:10.3390/toxins14110765_

Round 1

Reviewer 1 Report

The authors performed an interesting study of the hepatotoxic effects (and mode of action) of the extracts of Eupatorium fortunei, a medicinal plant used in China and other Asian contries. Despite being interesting, the study needs some corrections before its acceptance. All my comments and corrections can be find in the attached PDF. 

Author Response

Thank you very much for your patient reviewing work and helpful suggestions. We have carefully considered these suggestions, responded to them point-by-point, and revised the manuscript accordingly. Please see the attachment.

Reviewer 2 Report

The authors investigated the toxic mechanism induced by alkaloids in Eupatorium fortunei. 

The title should be revised, for example, as below: 

"Integrative Metabonomics and Proteomics Detected Hepatotoxicity in Mice Associated with Alkaloids from Eupatorium fortunei Turcz."

The English language needs critical editing by someone expert in academic writing. There are many poorly constructed sentences, grammatical mistakes, and poor word choices throughout the manuscript. For example, but limited to ... (I'm randomly picking a few sentences as an example).

L8-The aim of this study was to investigate the hepatotoxicity of total alkaloids in E. fortunei (EFTAs) and identified its toxic mechanisms to hepatocytes

L79: The organs of heart, liver, spleen, lung and kidney were separated from mice. Actually, 25 mg/kg/d EFTAs exposition for 4 weeks didn't produce obvious appearance change of various organs (Fig. 2A).

L238: Already stated earlier in the manuscript..... "we investigated the hepatotoxicity of EFTAs and identified its toxic 238 mechanisms to hepatocytes 

Figure 3. a, b, c, recheck the p values. I don't think the p values would be that significant. Also, cite p values in the text. 

The name of the plant and genes should be in italic font throughout the text. 

Define all the abbreviations separately as a table or separate section. 

Why did the authors choose 25 mg/kg/day EFTAs? Isn't it high for mice?  

Reference should be formatted as per the journal format—the year in bold. 

Even though the authors have done a detailed analysis, I would suggest getting the manuscript proofread by someone expert. 

Author Response

(The authors gave the same response as above.)

Reviewer 3 Report

This is a mostly well-designed study that addresses a relevant question. The experimental design is adequate and allows the drawing of the conclusions highlighted.

I have some concerns about this work, the most serious being:

A total of 20 animals were used, 10 for control and 10 for treatment conditions. Why? The authors do not provide any statistical analysis to justify their choice of number of animals. Why not 9? 8? This must be determined using power analysis.

Other relevant concerns are:

The chromatographic profile presented in Fig 1A is surprinsingly “clean”. Weren’t other compounds presente in fraction 2 apart from the 8 alkaloids?

Microscopy images (DAPI and TUNEL) are not clear and must be replaced by higher resolution/magnification ones. The scale bars are also impossible to see. How were apoptotic cells counted?

I have doubts regarding statistical analysis. In Figure 3, values for Con and EFTAs are largely coincidente and there is still a “*”. How can this be?

Densitometric quantification must be added to Figure 6B. Section 5.9 states that “The relative intensities of proteins were cal- 398 culated by Image J software.”, however they are nowhere to be found.

Information regarding extraction is insufficient: "Dried herbs (30 kg) of E. fortunei were cut and extracted by refluxing with 10 times of 50% (v/v) ethanol." The number of extractions (10) is not informative if the authors do not present how long each extraction took.

Statistical information is incomplete: Was the data normally distributed? Were outliers removed? How?

Author Response

(The authors gave the same response as above.)

Round 2

Reviewer 2 Report

The authors have answered most of my comments. I have a few minor comments. 

L67:  Write plant name in italic 

L177:  Contro correct it as Control 

L190: ...is not insignificant? Do you mean it is not significant 

L232: plant name in italic 

Author Response

(The authors gave the same response as above.)

Reviewer 3 Report

I understand that the authors based their choice regarding the number of animals in previously published papers, which varied from n=3 to n=15. However, previous papers are not enough, as there must be statistical and experimental reasons for the choice. What if a previous work wrongfully used n=50? This would not be ethical.

The criterium guiding the choice of number of animals should be power analysis, as preconized by International guidelines on animal wellfare. As so, the method claimed by the authors is not acceptable per si.

Statistics: We seem to have a problem here. The authors stated in their reply that “In the animal and cell experiments with limited sample size, the data are always non-Gaussian distribution3,4. Therefore, we didn’t pursue the normally distribution of our animal experiments”.

First of all, I fully disagree with this claim, as it is quite frequent that experiments with cells or animals be normally distrubted. The authors do not need to provide papers with this claim when verification is as simple and easy as conduct normality tests.

However, the most serious matter is that the authors state that that is non-Gaussian but still use t-test in all experiments, which is a a parametric test that assumes the distribution of the data. As so, the statistical tests used are inadequate and must be shifted towards a non-parametric test, such as Mann-Whitney test. This is an important matter that must be addressed.

Author Response

(The authors gave the same response as above.)
